# CCL5’s Role in Periodontal Disease: A Narrative Review

**DOI:** 10.3390/ijms242417332

**Published:** 2023-12-11

**Authors:** Katarzyna Barczak, Agnieszka Droździk, Mateusz Bosiacki, Ryta Łagocka, Diana Cenariu, Willi Andrei Uriciuc, Irena Baranowska-Bosiacka

**Affiliations:** 1Department of Conservative Dentistry and Endodontics, Pomeranian Medical University, Powstańców Wlkp. 72, 70-111 Szczecin, Poland; ryta.lagocka@pum.edu.pl; 2Laboratory of Preclinical Periodontology, Pomeranian Medical University, Powstańców Wlkp. 72, 70-111 Szczecin, Poland; agnieszka.drozdzik@pum.edu.pl; 3Department of Biochemistry and Medical Chemistry, Pomeranian Medical University in Szczecin, Powstańców Wlkp. 72, 70-111 Szczecin, Poland; mateusz.bosiacki@pum.edu.pl (M.B.); irena.baranowska.bosiacka@pum.edu.pl (I.B.-B.); 4MEDFUTURE—Research Center for Advanced Medicine, “Iuliu Hatieganu” University of Medicine and Pharmacy, 400337 Cluj-Napoca, Romania; diana.cenariu@umfcluj.ro; 5Faculty of Dental Medicine, “Iuliu-Hatieganu” University of Medicine and Pharmacy, 400012 Cluj-Napoca, Romania; willi.uriciuc@umfcluj.ro

**Keywords:** chemokine CCL5, periodontal diseases, gingival crevicular fluid, periodontitis, gingivitis, inflammation

## Abstract

Persistent host inflammatory and immune responses to biofilm play a critical role in the mechanisms that govern soft and hard tissue destruction in periodontal disease. Among the less explored facets of these mechanisms are chemokines, including CCL5 (C-C motif chemokine ligand 5), also known as RANTES (regulated on activation, normal T cell expressed and secreted), a proinflammatory CC subfamily chemokine synthesized by T lymphocytes. Despite its importance, there is currently no comprehensive review of the role of CCL5 in periodontitis in the literature. Therefore, this paper aims to fill this gap by summarizing the existing knowledge on the involvement of CCL5 in the onset and progression of periodontitis. In addition, we aim to stimulate interest in this relatively overlooked factor among periodontitis researchers, potentially accelerating the development of drugs targeting CCL5 or its receptors. The review examines the association of CCL5 with periodontitis risk factors, including aging, cigarette smoking, diabetes, and obesity. It discusses the involvement of CCL5 in pathological processes during periodontitis, such as connective tissue and bone destruction. The data show that CCL5 expression is observed in affected gums and gingival crevicular fluid of periodontitis patients, with bacterial activity contributing significantly to this increase, but the reviewed studies of the association between CCL5 expression and periodontal disease have yielded inconclusive results. Although CCL5 has been implicated in the pathomechanism of periodontitis, a comprehensive understanding of its molecular mechanisms and significance remains elusive, hindering the development of drugs targeting this chemokine or its receptors.

## 1. Introduction

Periodontal disease, encompassing gingivitis and periodontitis, is a persistent inflammatory condition that results in the degradation of the tissues supporting and surrounding the teeth [1]. When disease progresses, the loss of both the periodontal ligament and alveolar bone is obvious, ultimately leading to tooth loss which can impact mastication, aesthetics, patient self-assurance, and overall quality of life [2]. Furthermore, there exists compelling evidence of a potential connection between severe periodontitis and a minimum of 43 systemic diseases [1]. Strong association of periodontitis with some of these diseases can be attributed in part to systemic leakage of local inflammatory mediators and distribution of periodontal pathogens [3]. In cardiovascular diseases (CVD), periodontal pathogens leaking into blood vessels initiate the host’s immune response, which favors atheroma formation and maturation. *P. gingivalis, T. forsythia, A. actinomycetemcomitans*, and *Pr. intermedia* have been found in atheromatous plaques [4]. Indirect effects of bacteria predominantly involve host-mediated local and systemic immune responses expressed by increased circulating levels of proinflammatory cytokines and C-reactive protein (CRP) [5]. CRP, an inflammatory marker known to be associated with CVD is also found to be elevated in patients with chronic and aggressive periodontitis [6]. More aggressive forms of periodontitis presented a 50% greater rise in CRP levels in serum. The relationship between periodontitis and CRP has received great attention due to a link between periodontitis and systemic diseases including cardiovascular disease [7] and because of periodontal treatment effects on the level of CRP. According to current findings, based on the systematic review information, treatment of periodontitis induces a short-term acute inflammatory response increase when performed in an intensive session, whilst a progressive reduction up to 6 months is observed when treatment was performed in multiple sessions [6]. A later study confirmed that scaling and root planning treatment reduced serum CRP levels significantly [8].

The co-occurrence of periodontitis and systemic diseases elevates the significance of preventive measures, diagnostics, and treatment for periodontal disease, as well as of understanding molecular mechanism behind the pathogenesis of periodontitis involving inflammatory processes.

According to the Global Burden of Disease Study in 2016, advanced periodontal disease ranked as the 11th-most prevalent ailment worldwide [9]. Within the adult population (aged 35–44 years), periodontitis occurs in varying percentages ranging from 53% to 76% across different countries. In older demographics (aged 65–74 years), the prevalence escalates further, spanning from 53% to 100% [10].

Periodontitis stands as a multifaceted infectious ailment entailing numerous predisposing and etiological factors [1]. Beyond the requisite presence of oral biofilm for disease development [11], patients typically possess multiple risk factors for pathogenesis [12]. These risk factors can be categorized into non-modifiable and modifiable categories. Non-modifiable factors encompass age, gender, genetics, and ethnicity, while modifiable factors encompass oral hygiene, local risk factors, smoking, and systemic diseases [13]. Bouchard et al. introduced the concept of dividing periodontal risk factors into three groups: true risk factors with the strongest associations supported by extensive research, risk indicators, and predisposing factors based on limited evidence that does not permit a clear assessment of causality [11] (see Figure 1). The management of periodontal risk factors plays a pivotal role in the prevention and treatment of periodontal disease [14,15].

Although dental biofilms are undoubtedly a major cause of gingivitis, the connective tissue and bone destruction typical of periodontitis is driven by persistent host inflammatory and immune responses [14,15,16]. These responses to pathogens play a key role in the progression of periodontitis and explain treatment failure despite biofilm control in susceptible individuals [17].

Many studies show that the inflammatory response and the resident microbiome are linked in a bidirectional balance in health and gingivitis and in a bidirectional imbalance in periodontitis [14]. Subgingival plaque in health comprises an abundance of Gram-positive pathogens but some Gram-negative pathogens, such as *F. nucleatum, Veillonella* spp., and *Capnocytophaga* spp., are important components. Shifts observed in the development of gingivitis comprise an increase in bacterial biomass and change in its composition, with reduction of Gram-positive species such as *R. dentocariosa* and enrichment of Gram-negative species such as *F. nucleatum ss*, *Prevotella* spp., and *Selenomonas* spp. Likewise, development of periodontitis is accompanied by a huge increase in bacterial biomass and increased diversity in the composition of biofilm. Bacteria classified in the red complex (*T. denticola*, *P. gingivalis*, and *Tanneralla forsythia*) become increasingly dominant in the dysbiotic microbiota; several other species, like *Treponema spp*. and spirochaetes, are present. In addition to periopathogens, health-associated species are still present in periodontitis. Proportions of some species, referred to as core species, do not change from health to disease. Two of the most commonly detected species in this group are *Campylobacter gracilis* and *F. nucleatum ss. Vincentii* [14].

As in periodontitis, the presence of a biofilm containing pathogens conditions initiation of the inflammation in peri-implant tissues. It was believed that the peri-implant microbiota is similar to the periodontal microbiota in health or disease; however, with the advent of newer molecular techniques, it has become apparent that the peri-implant microbiota differ from periodontal microbiota. The peri-implant microbiota appear to be quantitatively lower and qualitatively higher for some bacteria genera [15]. At the peri-implant site, bacterial genera that are not present at the periodontal site, e.g., the genera *Burkholderia, Anaerovorax, Anaerococcus, Aerofilium*, and *Exiguobacterium*, are present, along with additional pathogens such as *Peptostreptococcus* spp. or *Staphylococcus epidermidis* and Staphylococcus aureus. The progression from the healthy peri-implant tissue to peri-implantitis shows changes in the composition of the microbiota in the absence of specific disease-causing bacteria.

A new concept called “inflammation-mediated polymicrobial-emergence and dysbiotic-exacerbation” (IMPEDE) attempts to explain how inflammation mediates biofilm dysbiosis and how its control influences dysbiosis [17]. This hypothesis is supported by results from animal studies showing that reducing inflammation prevents periodontal tissue destruction and modifies dysbiosis [18]. There are also promising results from clinical trials using microbiota-modulated agents as probiotics [19], parabiotics [20], postbiotics [21], and ozonated gels [22]. Topical antimicrobials supporting non-surgical mechanical therapy in treatment of periodontal disease, increased chances of successful treatment by reducing bacterial load and bacterial recolonization [23]. Adjuvant tools to mechanical oral hygiene procedures containing “-biotics” are likely to modulate the host immune system. The antioxidative modulation of probiotic bacteria involves a series of mechanisms [24]. It has been suggested that probiotics not only reduce bacterial counts due to competitive inhibition of adhesion sites but also exert an immunomodulatory response [25]. Paraprobiotics (heat-inactivated microbial cells) can regulate adaptive and innate immune systems thanks to their anti-inflammatory, antiproliferative, and antioxidant actions [26,27]. Postbiotics (concentrated bacterial active metabolites) have pleiotropic effects, including antiseptic, antioxidant, and immunomodulating effects [27]. The latter were shown to be as effective as chlorhexidine in reducing inflammatory indices [27].

Periodontal ligament destruction and alveolar bone resorption are primarily associated with dysregulation of the innate and adaptive immune systems [28,29].

The general molecular mechanism behind the pathogenesis of periodontitis has been extensively studied. However, the importance of lesser-known factors in the development of this disease has been relatively neglected. As a result, there are significant gaps in our understanding of the processes occurring in the periodontal tissues of individuals with periodontal disease. This knowledge gap hinders the development of novel drugs aimed at inhibiting the progression of this disease by targeting the lesser-understood factors.

Among these less explored factors in the molecular mechanisms of periodontal disease are chemokines, which function as chemotactic cytokines. In humans, nearly 50 members of this chemokine family have been identified [30]. These chemokines are divided into four subfamilies based on a conserved cysteine motif at their N-terminus; one of these subfamilies is the CC chemokines, which are characterized by a double cysteine motif. In humans, there are 24 representatives of CC chemokines, numbered from 1 to 28. Notably, there are also four CC chemokines that are present in mice but not in humans. Only a subset of these 24 CC chemokines are important in the pathogenesis of periodontitis, and CCL5 is one of them.

CCL5, first described in the 1980s [31], was first identified for its secretion by activated T lymphocytes. This led to the acronym “regulated upon activation, normally T expressed, and presumably secreted” (RANTES). CCL5 is a relatively small molecule with a mass of 8 kDa and consisting of 68 amino acid residues. It contains four cysteine residues, including two at its N-terminus, which classifies it as a member of the CC chemokine subfamily. The primary receptor for CCL5 is CCR5 [32,33], although CCL5 can also activate other CC chemokine receptors such as CCR1 [34,35], CCR3 [36], and CCR4 [35]. Notably, DPPIV/CD26 can modify CCL5 by removing the first two amino acids from its N-terminus, thereby increasing its affinity for CCR5 while decreasing its binding to other receptors, specifically, CCR1 and CCR3 [37]. This enzymatic modification may contribute to the specificity of the chemokine.

Due to the widespread expression of CCL5 receptors, this chemokine acts as a chemoattractant for various immune cell types, including monocytes [38], mast cells [35], dendritic cells [39], NK cells [40], eosinophils [41], basophils [42], CD4 T cells [43], CD8 T cells [44], and B cells [45].

To date, there has been no review in the literature summarizing the importance of the chemokine CCL5 in the molecular mechanisms of periodontitis. Therefore, the purpose of this paper is to collect and summarize the available knowledge about the role of this chemokine in the onset and development of periodontitis. Our aim is also to stimulate interest in this relatively unknown factor among researchers studying periodontitis. This may help to accelerate the research and development of new drugs against this disease by targeting this chemokine or the receptors for CCL5.

## 2. Results

### 2.1. CCL5 in Periodontitis

#### 2.1.1. CCL5 in the Gingival Crevicular Fluid and Gingiva of Patients with Periodontitis

The available literature contains the results of many studies on CCL5 in the gingiva of patients with periodontitis, usually showing similar results, but with some notable discrepancies. For example, the concentration of CCL5 in gingival crevicular fluid (GCF) has generally been shown to be higher in patients with periodontitis than in healthy subjects [46,47,48] and higher in affected areas than in unaffected areas [48]. However, there are also reports that CCL5 in GCF is not significantly different between patients with periodontitis—including aggressive periodontitis and severe forms of chronic periodontitis [49,50]—and healthy subjects; between sites affected by generalized periodontitis and healthy sites [51]; and between rheumatoid arthritis patients with periodontitis compared to those without periodontitis [52].

The total CCL5 in the GCF samples at one site was found to be higher in patients with chronic periodontitis than in healthy subjects [46,47,53,54]. In patients with generalized aggressive periodontitis, the total CCL5 in GCF was also higher than in healthy subjects and higher in progressive periodontitis (active sites) than in inactive sites [49]. However, some studies have found no statistically significant differences in CCL5 levels between patients with and without periodontitis [46,55] (Table 1, Figure 2).

Some studies have shown that CCL5 levels in GCF depend on the severity of periodontal disease. For example, Emingil et al. [49] showed that in patients with generalized aggressive periodontitis, the total CCL5 in the collected GCF per sample correlated positively with probing depth and clinical attachment loss. Another study showed that GCF collected from probing depths <3 mm in patients with periodontitis had lower CCL5 concentrations than GCF collected from probing depths >6 mm [46]. Although some studies show that CCL5 concentrations in GCF differed significantly between different probing depths, no difference was found when the total CCL5 in the total collected GCF per sample was analyzed, probably due to the different volume of GCF collected [46,47]. In another study, for the same amount of CCL5 in GCF, higher CCL5 concentrations were observed in GCF collected from shallower pockets [46].

The elevated CCL5 concentration in patients with periodontitis may be related to the inflammatory response in the gingiva. For this reason, the total CCL5 in the collected GCF per sample was correlated with the total amount of other cytokines, including IL-8 in GCF [55]. After treatment of periodontitis, CCL5 concentrations in GCF decreased and returned to levels observed in healthy subjects [47,51,55].

The results of studies analyzing gingival biopsies from patients with periodontitis show that CCL5 expression is higher in patients with periodontitis than in healthy subjects, confirming the results of GCF analyses [56].

#### 2.1.2. CCL5 in the Blood of Patients with Periodontitis

In a study of patients with periodontitis, gingival inflammation was associated with changes in plasma cytokine levels. Studies in patients with chronic periodontitis [57] and aggressive periodontitis [50] showed reduced blood levels of CCL5 compared to healthy subjects and reduced plasma levels of CCL11 and CXCL9 in patients with chronic periodontitis [57]. Plasma CCL5 in patients with chronic periodontitis was not significantly correlated with probing pocket depth, clinical attachment level, or bleeding on probing [57]. In contrast, patients with aggressive periodontitis were found to have lower levels of other cytokines, including CCL2, M-CSF, GM-CSF, IL-1ra, IL-10, and IL-13 [50]. At the same time, there are also studies suggesting that blood levels of CCL5 do not differ between healthy individuals and patients with chronic periodontitis [54].

#### 2.1.3. CCL5 in Periodontitis in the Elderly

Patient characteristics may be factors influencing CCL5 expression in periodontitis. Studies in rats have shown an increase in NF-*κ*B expression and an increase in CCL5 levels in aged gingival tissue [59]. This may indicate an increase in the inflammatory response in the gingiva during periodontal ageing and may explain the greater severity of periodontitis with advancing age in affected individuals [60]. In addition, a positive correlation between CCL5 levels in GCF and age has been observed in studies of patients with rheumatoid arthritis and periodontitis [52]. On the other hand, in a study by Gamonal et al., CCL5 levels in GCF were independent of patients’ age [46].

#### 2.1.4. CCL5 in Periodontitis and Obesity

The expression of CCL5 in the gingival crevicular fluid of patients with gingivitis may depend on the patient’s body mass index. The total amount and concentration of CCL5 in the GCF sample from patients with metabolic syndrome and gingivitis was higher than in lean individuals with gingivitis and obese individuals without gingivitis [61]. Lean individuals with gingivitis had higher CCL5 levels than lean individuals with healthy gums [61]. However, a study of rheumatoid arthritis patients with periodontitis showed that CCL5 levels in GCF were not correlated with the patients’ BMI [52] (Table 2).

#### 2.1.5. CCL5 in Periodontitis and Gender

Available studies report no differences in CCL5 levels in GCF according to patient gender [46,52].

#### 2.1.6. CCL5 in Periodontitis and Cigarette Smoking

The results of studies in patients with chronic periodontitis are inconsistent as to whether and how cigarette smoking affects CCL5 production in the gingiva of patients with this disease. Analysis of gingival tissue showed higher levels of CCL5 in smokers with chronic periodontitis compared to non-smokers with periodontitis [58]. In a study of patients with rheumatoid arthritis and periodontitis, CCL5 levels in GCF also correlated with cigarette smoking [52]. Another study showed that CCL5 concentrations in GCF were lower in smokers with periodontitis than in periodontitis patients who did not smoke cigarettes [48]. At the same time, smoking did not affect the total CCL5 in the collected GCF in the study by Haytural et al. [54]. The same study also showed that cigarette smoking did not affect serum CCL5 levels in patients with chronic periodontitis [54].

#### 2.1.7. CCL5 in Periodontitis and Diabetes

Epidemiological studies show that type 2 diabetes mellitus and periodontitis are related diseases and often occur together [63]. In addition, type 2 diabetes mellitus has been found to be associated with decreased CCL5 levels in patients with chronic periodontitis [62]. However, there has been no research that could clarify the causes and consequences of this.

#### 2.1.8. CCL5 in Periodontitis and Rheumatoid Arthritis

Rheumatoid arthritis may not affect CCL5 production in the gingiva of patients with periodontitis [52]. One study showed that patients with periodontitis and rheumatoid arthritis had higher levels of CCL5 in the GCF than patients with periodontitis but no rheumatoid arthritis, but the differences were not statistically significant (*p* = 0.087).

#### 2.1.9. CCL5 in Periodontitis and Vitamin D Deficiency

Vitamin deficiencies may influence the course of periodontitis. For example, vitamin D deficiency is associated with an increased risk of periodontal disease, which is partly related to an increased proinflammatory response, including the production of CCL5 in the gingiva [64].

### 2.2. Source of CCL5

#### 2.2.1. Cells Responsible for Increased Level of CCL5

The source of CCL5 in the gingiva of patients affected by periodontitis may be various cells that produce this chemokine in response to factors associated with this disease. Some of these factors are proinflammatory cytokines, in particular, IL-1*β*, as shown in a study using anakinra as an inhibitor [65]. The source of CCL5 itself may be fibroblasts; studies on gingival fibroblasts have shown that inflammatory responses, including the action of proinflammatory cytokines such as TNF-*α* and IL-1*β*, increase the expression of CCL5 in these cells and, hence, CCL5 secretion [66].

Monocytes may be another important cell involved in the production of CCL5 [67]. Patients with periodontitis have been shown to have elevated numbers of CD14^+^ monocytes in the GCF [67]*—*cells that demonstrate high expression of CCR2 and CCR5 compared to CD14^−^ cells in the GCF. This suggests that these monocytes are recruited by ligands of these receptors, in particular, CCL2, whose levels are elevated in GCF from patients with periodontitis [67]. The quoted data suggest that, in addition, CCL5 may also be involved in gingival infiltration by monocytes in periodontitis. CD14^+^ monocytes secrete a variety of cytokines, including TNF-*α*, IL-1*β*, IL-6, IL-8, CCL2, CCL3, and CCL5, compared to CD14^−^ cells from GCF in patients with periodontitis [67], which suggests that CD14^+^ monocytes may be a source of CCL5 in GCF and gingival tissue in patients with periodontitis.

#### 2.2.2. Bacteria as a Factor Responsible for Increased Level of CCL5

Bacteria are the factor responsible for the increased expression and production of CCL5 in gingival cells in patients with periodontitis. Studies of the HeLa cell line, U937 macrophages [68], a human whole blood model [69], bone marrow-derived macrophages [70], calvarial osteoblasts, and a bone marrow-derived stromal cell line ST2 [71] have shown that the presence of *Porphyromonas gingivalis* increases CCL5 production in these cells via a mechanism dependent on the activation of NF-*κ*B [71]. The bacterium also increases CCL5 production in gingival mesenchymal stromal cells (GMSCs) from patients with periodontitis [72], an effect not observed in healthy individuals.

Other studies have also shown that *Porphyromonas gingivalis* induces CCL5 expression. An in vitro study has also shown that *Porphyromonas gingivalis* haemagglutinin B increases CCL5 expression in dendritic cells and T cells [73]. The bacterium also increases the production of other proinflammatory cytokines such as IL-1*β*, TNF-*α*, IL-6, IFN-*γ*, CCL2, CCL3, PGE_2_, and CXCR2 ligands [69,70]. However, studies in gingival epithelial cells of the HEp-2 line showed that *Porphyromonas gingivalis* did not increase the expression of CCL5 in these cells whereas it increased the expression of IL-1*β*, IL-8, and ICAM-1 [73,74]. In an in vivo study on mice, oral infection with *Porphyromonas gingivalis* was shown to increase plasma CCL5 levels [75]. However, *Porphyromonas gingivalis* infection does not appear to significantly affect CCL5 production in gingiva affected by periodontitis in humans. A study of patients with generalized chronic periodontitis showed that the ratio of Porphyromonas gingivalis in plaque did not significantly correlate with the amount of CCL5 in GCF [53].

A study on human periodontal fibroblasts showed that *Fusobacterium nucleatum* increased the expression of CCL5 and CCL2, CXCL1 at the mRNA, and protein levels [56]. *Tannerella forsythia* also increased CCL5 production in the HeLa cell line and U937 macrophages at the protein level [68]. A study in mice showed that infection with *Aggregatibacter (Actinobacilus) actinomycetemcomitans* increases CCL5 production in the gingiva [76]. *Aggregatibacter (Actinobacilus) actinomycetemcomitans* has also been shown to induce an inflammatory response, as shown in a study on Raw 264.7 macrophages [77] and CD4^+^ T lymphocytes [78]. In these cells, the bacteria increased the expression of CCL5. *Actinobacillus actinoinycetemcomitans* increased the expression of CCL3, CCL4, CCL5, CXCL2, and CXCL10. A study in mice also showed that infection with *Actinobacillus actinomycetemcomitans* led to an inflammatory response in the gingiva [79]. Infection with this bacterium was shown to increase the production of IFN-*γ*, TNF-*α*, and chemokines such as CCL3, CCL4, CCL5, CCL11, and CXCL1. This was followed by the infiltration of gingiva infected with this bacterium by cells of the immune system, including cells expressing the receptor for CCL5: CCR5 [79].

Not all bacteria that cause periodontitis increase the expression or production of CCL5. *Treponema denticola* reduces CCL5 production in the HeLa cell line and in U937 macrophages [68], which is related to the action of bacterial proteases that degrade CCL5; a similar effect has been shown for *Porphyromonas gingivalis* [68,80]. It is not only CCL5 that is degraded by bacterial proteases; other cytokines, such as IL-6 and IL-8, are also degraded in the same way [80].

## 3. Discussion

### 3.1. CCL5 in Disease Mechanisms of Periodontitis

#### 3.1.1. CCL5-CCR5 Axis Gene Polymorphism as an Indicator of CCL5 Significance in the Pathogenesis of Periodontitis

CCL5 is not only a marker for periodontitis, it is also involved in the pathological processes of this disease, as confirmed by the results of research on the CCL5 gene polymorphisms. A study of the Taiwanese population showed that the AA and AG genotype—403 single nucleotide polymorphism of chemokine ligand 5 (CCL5-403)—is associated with a higher incidence of generalized aggressive periodontitis [81]. At the same time, a 32 bp deletion of CCR5 (CCR5Δ32) was not associated with an increased likelihood of developing periodontitis [81]. CCL5-403 is a polymorphism affecting the promoter of the CCL5 gene [81,82], and CCL5-403 genotype A is associated with increased CCL5 promoter activity, presumably through the creation of a new binding site for a transcription factor [82]. For this reason, it is reasonable to assume that the greater the synthesis of CCL5, the greater the likelihood of developing periodontitis. However, a study on British and Swedish patients has shown that CCR5Δ32 polymorphisms and CCL5 A/G polymorphisms are not correlated with the incidence of chronic periodontitis [83].

#### 3.1.2. Involvement of CCL5 in Gingival Infiltration by Immune Cells in Periodontitis

The most important property of chemokines is the chemoattraction of immune cells that highly express the receptor for a particular chemokine. Studies of the gingiva in periodontitis have shown that periodontitis patients are infiltrated by cells expressing the receptor CCR5 [47,76,84], and that cells with CCR5 expression are not observed in healthy gingiva [84]. The main immune cells expressing this receptor and infiltrating gingival tissue may include Th1 cells and macrophages [76], although the exact role of CCL5 in the recruitment of these cells has not been investigated in models of periodontitis. It appears that either this chemokine or another CCR5 ligand may play an important role in this process.

#### 3.1.3. CCL5 as a Factor in Alveolar Bone Destruction

A study involving a mouse model of periodontitis induced by infection with *Aggregatibacter actinomycetemcomitans* showed that CCR1 and CCR5 are important in alveolar bone resorption [76]. However, whether CCL5 or another chemokine plays an important role in this process has not been confirmed. Based on the analysis of chemokine expression, it can be speculated that CCL4 and CCL5 may be important in this process [76], although the direct mechanism is not fully understood.

CCL5 may act directly on osteoclasts, which, just like osteoclast precursors, express CCR1 [85,86]. The expression of this receptor is increased during the differentiation of precursors into osteoclasts [86]. Therefore, osteoclasts respond to ligands for CCR1, in particular, for CCL3, CCL5, and CCL23 [85]. Studies on CCL5 have shown that this chemokine can stimulate osteoclast precursors to fuse [86], induce osteoclast chemotaxis [85], and increase the speed of chemotaxis of these cells [87], resulting in increased osteoclast activity at sites of inflammatory response. However, in periodontitis, other factors appear to be responsible for osteoclast migration that are more effective at lower concentrations on these cells [87]. Examples include CCL2, CCL3, CXCL12, and C5a [87]. Some studies show that CCL5 can also act as a chemoattractant for osteoblasts [88] and can increase the viability of these cells. Therefore, CCL5 may increase osteoblast activity, especially through its receptor, CCR5 [89]. This property suggests that CCL5 may inhibit bone degeneration in periodontitis, although further research is needed to investigate the exact effect of CCL5 on bone tissue in periodontitis (Table 3).

#### 3.1.4. Effect of CCL5 on Periodontal Ligament Stem Cells

In periodontitis, CCL5 may not contribute to the development of the disease but instead may inhibit its progression. It may inhibit the development of periodontitis by acting on periodontal ligament stem cells (PDLSCs) [90]. The induction of PDLSC migration by CCL5 results in PDLSC homing to sites involved in inflammatory responses, including those in periodontitis [90]. PDLSCs secrete factors that have regenerative effects on lesions caused by periodontitis [91].

#### 3.1.5. Link between Periodontitis and Cancer Progression: Impact on Lymph Node Metastasis

Periodontitis is not only a disease of the periodontium, but it also influences systemic health in which CCL5 plays an important role. Whole blood cell cultures from patients with periodontitis show a hyper-responsiveness to LPS compared to whole blood cells from healthy individuals without periodontitis [92]. These cells from people with periodontitis produce more CCL5 but less IL-12p70 in response to LPS, which suggests that the immune response of patients with periodontitis differs from that of patients without periodontitis. This may have implications for other inflammatory diseases, including cancer. In patients with periodontitis, gingival disease facilitates lymph node metastasis, particularly in breast cancer, which is associated with increased expression of CCL5, CCL2, and CXCL5 [65]. These chemokines induce the recruitment of macrophages and granulocytic MDSCs into the cervical lymph nodes. As a result, these cells form a pre-metastatic niche that facilitates metastasis in these lymph nodes [65].

## 4. Materials and Methods

The PubMed search engine (https://pubmed.ncbi.nlm.nih.gov/ accessed on 20 August 2023) was used to search for experimental articles on the importance of CCL5 in periodontal disease (gingivitis and periodontitis). The phrase (“periodontal diseases” or “periodontitis”) (CCL5 or RANTES) not review was used to search for the given articles.

Articles were searched without any time restrictions on when they were published. They had either of two specific terms in the title, keywords, or abstract. The first was the name of the chemokine in question, CCL5. The early name for this chemokine, RANTES, was also included in the search. As in the 2000s, both names were used interchangeably; the phrase (CCL5 or RANTES) was used to search for all articles on this chemokine. The second term searched for in articles was periodontitis. This phrase refers to the importance of the CCL5 chemokine in periodontitis, so articles were searched for that included the name of the disease periodontitis in addition to the name of the CCL5 chemokine. As periodontitis is one of the periodontal diseases, the phrase (“periodontal diseases” or periodontitis) was used to search for articles that described periodontal diseases but did not contain the word periodontitis in the title, keywords, or abstract. Due to the fact that the written article is a review paper, only experimental papers were taken into consideration, and review papers were rejected. Therefore, the phrase not review was used in the PubMed browser.

In total, 136 articles were found in the PubMed search engine. Based on the title and abstract, 49 of them were selected as containing chapters on the importance of CCL5 in periodontitis.

## 5. Conclusions

CCL5 chemokine expression in periodontitis patients is elevated in the affected gums and gingival crevicular fluid, and bacterial activity is a significant factor in increasing CCL5 expression in periodontitis patients.

The results of studies on the association of CCL5 expression with periodontal disease depending on patient risk factors such as age, obesity, smoking, and diabetes are inconclusive.

CCL5 is involved in the pathomechanism of periodontitis; however, studies on the molecular mechanisms and significance of CCL5 in periodontitis are insufficient to understand the action of CCL5 in this disease and to develop drugs targeting this chemokine or its receptors.

## Figures and Tables

**Figure 1 ijms-24-17332-f001:**
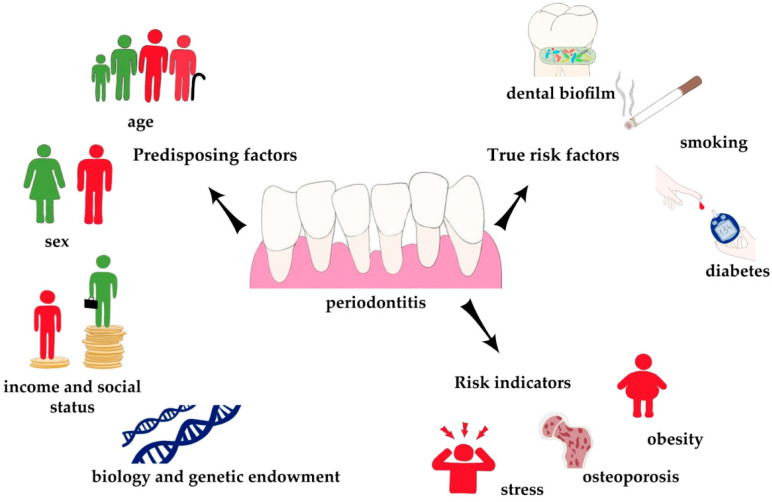
Risk factors for periodontitis.

**Figure 2 ijms-24-17332-f002:**
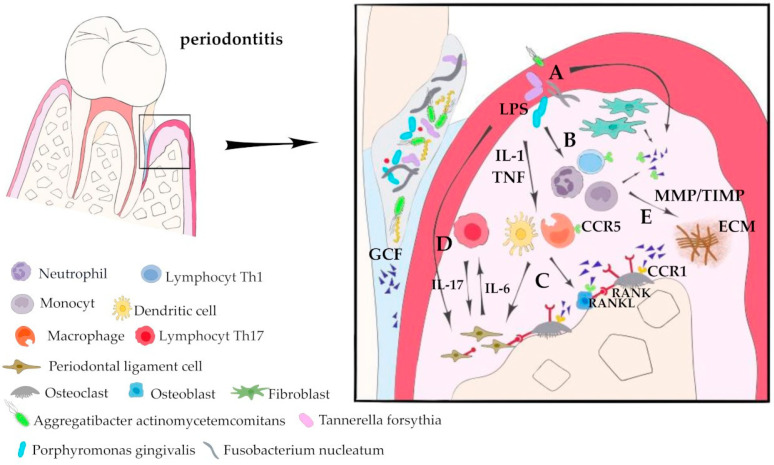
Involvement of CCL5 in pathological processes in periodontitis. Dysbiotic bacteria spreading over the inflamed gingiva (**A**), together with its component lipopolysaccharide (LPS), lead to the initial inflammatory events expressed by leukocyte recruitment into the periodontal space (**B**). The first line of immune response, i.e., macrophages and dendritic cells, produce IL-1 and TNF, which, in turn, stimulate periodontal ligament cells and osteoclasts to increase RANKL, create an imbalance in osteoprotegerin (OPG) levels, and promote bone resorption (**C**). LPS-stimulated periodontal ligament cells release IL-6, which induces the accumulation of Th17 cells (**D**). These cells, through the production of IL-17, induce RANKL from osteoclasts and periodontal ligament cells, thereby stimulating osteoclastic bone resorption. The local chronic inflammatory response also leads to an imbalance in the MMP/TIMP ratio, which is responsible for the destruction of the extracellular matrix (ECM) of soft and mineralized periodontal tissues (**E**).

**Table 1 ijms-24-17332-t001:** CCL5 analyses in patients with periodontitis and gingivitis.

Source	Result of the Analysis	Number of Patients/Number of Healthy People in the Study	Method of Analysis/Type of Sample
Gamonal et al., 2000 [46]	Higher concentration and total CCL5 in GCF in patients with periodontitis. CCL5 concentration is higher in GCF at sites with <3 mm probing depth than in >6 mm, but total CCL5 is at the same level regardless of probing depth.	12 people with periodontitis; 12 healthy subjects	ELISA; GCF
Gamonal et al., 2001 [47]	Higher CCL5 concentrations in patients with periodontitis. Quantity and concentration are not significantly associated with probing depth.	12 people with periodontitis; 6 healthy subjects	ELISA; GCF
Tymkiw et al., 2011 [48]	CCL5 concentration is higher in sites affected by periodontitis than in sites not affected by periodontitis. CCL5 concentration is higher in sites covered by periodontitis compared to samples from healthy subjects. In smokers, reduced CCL5 concentration in sites covered and not covered by periodontitis.	40 patients with periodontitis (20 smokers, 20 non-smokers); 12 healthy subjects	multiplexed fluorescent bead-based immunoassay; GCF
Emingil et al., 2004 [49]	Higher total CCL5 per sample in patients with generalized aggressive periodontitis than in healthy subjects; concentrations did not differ between patients and healthy subjects. The amount of CCL5 positively correlated with probing depth and clinical attachment loss.	20 people with periodontitis; 10 healthy subjects	Enzyme immunoassay; GCF
Zein Elabdeen et al., 2017 [50]	CCL5 levels are reduced in patients with periodontitis compared to healthy subjects. CCL5 levels in GCF do not differ between patients with periodontitis and healthy subjects.	19 people with aggressive periodontitis; 19 healthy subjects	multiplex fluorescent bead-based immunoassays; plasma
Thunell et al., 2010 [51]	The concentration of CCL5 in sites affected by periodontitis does not differ from non-periodontitis sites. After therapy, CCL5 concentration decreases.	6 patients with generalized severe chronic periodontitis	multiplex fluorescent bead-based immunoassay; GCF
Söderlin et al., 2021 [52]	CCL5 concentration is higher but not statistically significant (*p* = 0.087) in patients with periodontitis and with rheumatoid arthritis compared to patients with rheumatoid arthritis but without periodontitis. CCL5 concentration positively correlated with patient age and smoking but not with gender and BMI.	80 people with periodontitis and rheumatoid arthritis; 52 people with rheumatoid arthritis but without periodontitis	xMap technology (bead-based multiplexed immunoassay); GCF
Shimada et al., 2013 [53]	Higher total CCL5 in periodontitis sites than in non-periodontitis sites.	11 patients with generalized chronic periodontitis	multiplexed fluorescent bead-based immunoassay; GCF
Haytural et al., 2015 [54]	Total CCL5 in GCF is elevated in patients with periodontitis relative to healthy subjects. Smoking does not affect the total CCL5 in GCF.	30 patients with chronic periodontitis (20 smokers) and 20 healthy subjects (10 smokers)	multiplex immunoassay; GCF
Haytural et al., 2015 [54]	Serum CCL5 concentrations in patients with periodontitis do not differ from healthy subjects. Smoking does not affect serum CCL5 concentrations.	30 patients with chronic periodontitis (20 smokers) and 20 healthy subjects (10 smokers)	multiplex immunoassay; serum
Gamonal et al., 2000 [55]	CCL5 concentration is higher in GCF at sites with <3 mm probing depth than in >6 mm, but total CCL5 is at the same level regardless of probing depth.	12 people with periodontitis; 12 healthy subjects	ELISA; GCF
Rath-Deschner et al., 2020 [56]	Higher expression of CCL5 at the mRNA level in people with periodontitis.	6 people with periodontitis; 6 samples from healthy sites	qRT-PCR; gingival biopsies
de Queiroz et al. 2008 [57]	CCL5 levels are reduced in patients with periodontitis relative to healthy subjects.	17 people with chronic periodontitis; 8 healthy subjects	Inflammatory flow cytometry multiplex assays; serum
Souto et al., 2014 [58]	Higher CCL5 concentrations in smokers than in non-smokers.	43 patients with chronic periodontitis (23 smokers, 20 non-smokers)	ELISA; gingival tissues

**Table 2 ijms-24-17332-t002:** CCL5 levels in periodontal diseases in relation to patient characteristics and other diseases.

Description	Characteristic
Inflammatory responses are increased in older adults. In subjects with rheumatoid arthritis and periodontitis, CCL5 levels correlated positively with age [52]. Other studies have not confirmed this [46].	Age
CCL5 levels are higher in obese patients with gingivitis than in lean patients with and without gingivitis [61]. Other studies have not confirmed this [52].	Obesity
CCL5 levels not related to gender in periodontitis patients.	Sex
Higher CCL5 levels in smokers with periodontitis than in non-smokers with periodontitis [52,58]. The opposite has also been observed [48].	Cigarette smoking
Patients with diabetes and periodontitis have lower CCL5 levels than patients with diabetes alone [62].	Diabetes
Unrelated.	Rheumatoid arthritis

**Table 3 ijms-24-17332-t003:** Involvement of CCL5 in disease mechanisms of periodontitis.

Description	Mechanism
Potential recruitment of Th1 cells and macrophages by CCL5	Immune cell infiltration
Increased osteoclast activity (stimulation of osteoclast precursors to fuse, induction of osteoclast chemotaxis).	Alveolar bone destruction
CCL5 induces chemotaxis of PDLSCs and, thus, regenerative effects on lesions caused by periodontitis.	PDLSCs

## Data Availability

Not applicable.

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
