# Peer review of "CCL5’s Role in Periodontal Disease: A Narrative Review"

_ijms, 2023, doi:10.3390/ijms242417332_

Round 1

Reviewer 1 Report

Comments and Suggestions for Authors

Manuscript of considerable interest for the dental sector, requires a major revision before being evaluated for publication.

Abstract: very short and not very descriptive in the results part.

Keywords; few, to add specific ones registered on MeSH.

Introduction, the role of c-reactive protein in the progression of periodontal disease is missing, and the difference of the oral microbiota in periodontal patients in a state of tissue integrity, compared to inflammation and compared to implant patients (Scribante et al.)

The role of neutrophils is also missing, especially in heart disease patients which have not been included in this manuscript, recently published in this journal.

And as future objectives add the proactive role to reduce the incidence of inflammation, and how natural substances act, such as probiotics, paraprobiotics, postbiotics and ozonated substances in a reactive phase.

Add proactive action in your conclusions

Author Response

We would like to thank you for kind and insightful review of our manuscript. We have tried very carefully to respond to all comments and improve our manuscript.

Review 1

Comments and Suggestions for Authors

Manuscript of considerable interest for the dental sector, requires a major revision before being evaluated for publication. 

Abstract: very short and not very descriptive in the results part. 

According to Reviewer remark we corrected the abstract section

Keywords; few, to add specific ones registered on MeSH. 

According to Reviewer remark we corrected the keywords

Introduction, the role of c-reactive protein in the progression of periodontal disease is missing, and the difference of the oral microbiota in periodontal patients in a state of tissue integrity, compared to inflammation and compared to implant patients (Scribante et al.) 

According to Reviewer remark we corrected the Introduction section

The role of neutrophils is also missing, especially in heart disease patients which have not been included in this manuscript, recently published in this journal. 

And as future objectives add the proactive role to reduce the incidence of inflammation, and how natural substances act, such as probiotics, paraprobiotics, postbiotics and ozonated substances in a reactive phase. 

According to Reviewer remark we corrected this sections.

Add proactive action in your conclusions

According to Reviewer remark we corrected conclusions.

Reviewer 2 Report

Comments and Suggestions for Authors

The study is relevant and current. The role of CCL5 in periodontal disease has been widely investigated in recent years. However, the manuscript can be improved in several aspects. The manuscript needs to be restructured following the classic IMRDC scheme. A section with the methodology, another with the results and another with the discussion and conclusions should be included.

First, authors must specify what type of review they perform. In the abstract they indicate that "This review discusses CCL5 association with risk factors for perio-dontitis such as aging, smoking, diabetes, obesity as well as involvement of CCL5 in patho-logical processes in the course of periodontitis, connective tissue and bone destruction ", but in the Introduction they do not make clear what the rationale of this review is or its specific objective. The authors must add a paragraph at the end of the Introduction indicating why a review on CCL5 and periodontal disease is necessary and the type of review they propose to carry out (narrative, scoping, systematic...).

Additionally, the type of review should be included in the title.

Secondly, a Methodology section should be added explaining how the selection of the articles included in the review was carried out. The inclusion criteria, the time period...

Thirdly, it would be very useful for readers to include a summary table of everything discussed in the CCL5 and periodontitis section.

Finally, authors must include a Results and Discussion section, in which they include everything discussed in the manuscript. At the end, they must also include a section with the most relevant conclusions and the clinical importance that the result of the review could have.

Author Response

Review 2

We would like to thank you for kind and insightful review of our manuscript. We have tried very carefully to respond to all comments and improve our manuscript.

Comments and Suggestions for Authors

The study is relevant and current. The role of CCL5 in periodontal disease has been widely investigated in recent years. However, the manuscript can be improved in several aspects. The manuscript needs to be restructured following the classic IMRDC scheme. A section with the methodology, another with the results and another with the discussion and conclusions should be included.

First, authors must specify what type of review they perform. In the abstract they indicate that "This review discusses CCL5 association with risk factors for perio-dontitis such as aging, smoking, diabetes, obesity as well as involvement of CCL5 in patho-logical processes in the course of periodontitis, connective tissue and bone destruction ", but in the Introduction they do not make clear what the rationale of this review is or its specific objective. The authors must add a paragraph at the end of the Introduction indicating why a review on CCL5 and periodontal disease is necessary and the type of review they propose to carry out (narrative, scoping, systematic...).

According to Reviewer remark we corrected the Introduction section.

Additionally, the type of review should be included in the title.

According to Reviewer remark we corrected the title.

Secondly, a Methodology section should be added explaining how the selection of the articles included in the review was carried out. The inclusion criteria, the time period... 

According to Reviewer remark we corrected we methodology section.

Thirdly, it would be very useful for readers to include a summary table of everything discussed in the CCL5 and periodontitis section. 

According to Reviewer remark we corrected and added summary table.

Finally, authors must include a Results and Discussion section, in which they include everything discussed in the manuscript. At the end, they must also include a section with the most relevant conclusions and the clinical importance that the result of the review could have.

According to Reviewer remark we added and corrected the Discussion section.

Round 2

Reviewer 1 Report

Comments and Suggestions for Authors

The manuscript has been properly revised and can be published

Reviewer 2 Report

Comments and Suggestions for Authors

The authors have followed the suggestions of the reviewers point the manuscript, it has been substantially improved together in my opinion, it can now be accepted for publication.